# Cellular and Structural Changes in Achilles and Patellar Tendinopathies: A Pilot In Vivo Study

**DOI:** 10.3390/biomedicines12050995

**Published:** 2024-04-30

**Authors:** Dimitrios Kouroupis, Carlotta Perucca Orfei, Diego Correa, Giuseppe Talò, Francesca Libonati, Paola De Luca, Vincenzo Raffo, Thomas M. Best, Laura de Girolamo

**Affiliations:** 1Department of Orthopedics, UHealth Sports Medicine Institute, Miller School of Medicine, University of Miami, Miami, FL 33146, USA; dxk504@med.miami.edu (D.K.);; 2Diabetes Research Institute & Cell Transplant Center, Miller School of Medicine, University of Miami, Miami, FL 33136, USA; 3Laboratorio di Biotecnologie Applicate all’Ortopedia, IRCCS Istituto Ortopedico Galeazzi, Via C. Belgioioso 173, 20157 Milan, Italylaura.degirolamo@grupposandonato.it (L.d.G.); 4Cell and Tissue Engineering Laboratory, IRCCS Istituto Ortopedico Galeazzi, Via C. Belgioioso 173, 20157 Milan, Italy

**Keywords:** Achilles tendinopathy, patellar tendinopathy, tendon stem/progenitor cells, macrophages, immunomodulation

## Abstract

Tendinopathies continue to be a challenge for both patients and the medical teams providing care as no universal clinical practice guidelines have been established. In general, tendinopathies are typically characterized by prolonged, localized, activity-related pain with abnormalities in tissue composition, cellularity, and microstructure that may be observed on imaging or histology. In the lower limb, tendinopathies affecting the Achilles and the patellar tendons are the most common, showing a high incidence in athletic populations. Consistent diagnosis and management have been challenged by a lack of universal consensus on the pathophysiology and clinical presentation. Current management is primarily based on symptom relief and often consists of medications such as non-steroidal anti-inflammatories, injectable therapies, and exercise regimens that typically emphasize progressive eccentric loading of the affected structures. Implementing the knowledge of tendon stem/progenitor cells (TSPCs) and assessing their potential in enhancing tendon repair could fill an important gap in this regard. In the present pilot in vivo study, we have characterized the structural and cellular alterations that occur soon after tendon insult in models of both Achilles and patellar tendinopathy. Upon injury, CD146^+^ TSPCs are recruited from the interfascicular tendon matrix to the vicinity of the paratenon, whereas the observed reduction in M1 macrophage polarization is related to a greater abundance of reparative CD146^+^ TSPCs in situ. The robust TSPCs’ immunomodulatory effects on macrophages were also demonstrated in in vitro settings where TSPCs can effectively polarize M1 macrophages towards an anti-inflammatory therapeutic M2 phenotype. Although preliminary, our findings suggest CD146^+^ TSPCs as a key phenotype that could be explored in the development of targeted regenerative therapies for tendinopathies.

## 1. Introduction

The term tendinopathy is generally used to describe a spectrum of pathological changes affecting the tendons, involving pain, diffuse or localized fibrosis, loss of tissue integrity, and impaired performance. Tendinopathies are a common and significant clinical problem not only caused by sports injuries but also by aging [1,2,3], highly repetitive movements for occupational reasons [3], side effects of particular drug therapies [4], and metabolic disorders [1]. In the lower limb, tendinopathies affecting the Achilles and the patellar tendons are the most common, showing an incidence in athletic populations of 13% and 21%, respectively [5], compared to the general population, with and incidence of 2.4% and 1.6% for the Achilles and patellar tendinopathies, respectively [1]. 

To date, despite the high rate of incidence of tendinopathies, there are no universal consensus treatments to restore native function to the damaged tissue [6,7].

Currently, in tendinopathy management, several therapeutic approaches exist in clinical practice, including conservative treatments, pharmacological interventions, and therapeutic exercise [8], and other innovative approaches are still being studied, such as the use of decellularized scaffolds [9], but definitive solutions have not yet been obtained.

The paucity of effective therapeutic solutions may reflect a certain lack of understanding of tendon biology and related pathologies.

The case of inflammation is problematic; its role in the development and progression of tendinopathy has recently been revisited and defined as crucial, especially in the initial phase, i.e., when the tendinopathy is not yet clearly evident from a clinical point of view, although histologically there is often tissue compromise [10,11]. Some studies have highlighted the presence of inflammatory factors [12] and immune cells (macrophages, T cells, mast cells) [13,14,15,16] in the initial stages of the disease as a consequence of repeated trauma or mechanical stress of the tissue. The presence of these cellular components triggers catabolic responses, leading to aberrant matrix changes and therefore tissue degeneration [17]. Emerging evidence suggests that alarmins released from necrotic cells trigger a consequent inflammatory response and activation of the innate immune system [18]. Tenocytes together with resident (mast cell and M2 macrophages) and infiltrative (mast cells, T-cells, M1 macrophages) immune cells are key players in the stromal and immune sensing compartments involved in tendon pathophysiology [18]. Therefore, the massive release of inflammatory cytokines and the increase in matrix metalloproteases (MMP1 and MMP7) affect the reparation–degeneration milieu of the tissue, resulting in a state of protracted, dysregulated, and maladaptive response to damage [12,18,19]. The intrinsic healing potential of the tissue is limited and unable to counteract this inflammatory state; therefore, the pre-injury state is not always restored, and fibrotic changes coupled with impairment of mechanical and structural function often result [7,20].

The presence within the tendon of resident stem/progenitor cells (tendon stem/progenitor cells, TSPCs) [21] has motivated recent interest addressing tendon healing due to some similarities with gold-standard mesenchymal stromal cells of the bone marrow and adipose tissue in terms of their self-renewal, clonogenicity, multipotency, and immunophenotypic properties [16,22,23]. CD146 represents one of the markers identifying TSPCs. CD146^+^ TSPCs endowed with a paracrine role with high regenerative potential represent an exploitable key phenotype to innovative therapeutic approaches for tendinopathy [24]. Preliminary assessments in vitro suggest a possible involvement of TSPCs in the tissue’s immunoregulatory response [25]. Moreover, exosomes released by TSPCs can induce a change in macrophage phenotypic expression, resulting in an increase in anti-inflammatory factors (IL-10) and a decrease in inflammatory cytokines (IL-6) [26]. However, several aspects concerning TSPCs, their properties, and their identification both in vitro [25] and in vivo [27] are somewhat controversial or still unclear, especially with reference to their role during the early stages of tendinopathy.

For all these reasons, the aims of the current study are (1) to investigate the structural and cellular alterations in tendinopathies; and (2) to investigate the presence of TSPCs in healthy and pathological tendons and to evaluate their immunomodulatory ability.

## 2. Materials and Methods

### 2.1. Study Approval

This study was performed at the University of Miami (UM-Miami) and IRCCS Istituto Ortopedico Galeazzi (IOG-Milan) in accordance with the Declaration of Helsinki. All animal procedures were approved by the Institutional Animal Care and Use Committee at the University of Miami (approval 19-001-VVC ad01) and were in compliance with ARRIVE guidelines [28,29].

After approval of the protocol by the local IOG Institutional Review Board (M-SPER-014-Ver.8-08.11.2016) and the receipt of informed consent for the collection of waste material by the patients recruited at IOG, tendon tissues were collected. Following cell isolations, samples were transferred to UM-Miami in order to perform further analysis.

### 2.2. Tendon Stem/Progenitor Cells Isolation and Cultures

Semitendinosus and gracilis tendons were collected from donors (n = 6, males, 33 ± 8 y/o, caucasians) undergoing elective anterior cruciate ligament (ACL) reconstruction. Demographic data of donors are reported in Table 1.

Harvested tendons were enzymatically digested for 16 h with 0.3% *w*/*v* Collagenase type I (185 U/mg, Worthington Biochemical Corporation, Lakewood, NJ, USA) to clean the sample of the residues of other tissues and to isolate human tendon stem/progenitor cells (TSPCs). Enzymatic digestion was inactivated with complete media with DMEM low-glucose (1 g/L) GlutaMAX (ThermoFisher Scientific, Waltham, MA, USA) containing 10% fetal bovine serum (FBS; VWR, Radnor, PA, USA), washed, and seeded at a density of 1 × 10^6^ cells/175 cm^2^ flask in chemically reinforced (Ch-R) medium. Mesenchymal Stem Cell Growth Medium 2 was mixed with the provided supplements to obtain complete Ch-R medium, as reported in the manufacturer’s instructions (PromoCell, Heidelberg, Germany). A total of 48 h after seeding, the medium with non-adherent cells was replaced with a fresh medium. TSPCs were cultured at 37 °C, 5% (*v*/*v*) CO_2_, until 80% confluent as passage 0 (P0). Subsequently, they were detached with TrypLE™ Select Enzyme 1X (Gibco, ThermoFisher Scientific) and plated at 1:5 dilution to P3. At each passage, cell viability was assessed using 0.4% (*w*/*v*) Trypan Blue (Invitrogen, ThermoFisher Scientific).

### 2.3. Immunophenotypic Profiling

TSPCs (2 × 10^5^) were detached and stained for 20 min at 4 °C in the dark with fluorescently conjugated anti-human antibodies CD90-FITC (Clone 5E10, BioLegend, San Diego, CA, USA), CD105-PE (Clone SN6h, BioLegend), CD44-BV605 (Clone IM7, BioLegend), CD73-APC (Clone AD2, BioLegend), and CD146-PE (Clone 541-10B2, Miltenyi Biotec, Bergisch Gladbach, Germany) and analyzed using a CytoFLEX flow cytometer (Beckman Coulter Life Sciences, Brea, CA, USA). Collection of 50,000 events for each cell sample was performed. Subsequent gating strategies were standardized for each sample based on the following criteria: scatter, singlets, and positive expression. The resulting data were overlaid with corresponding isotype controls.

### 2.4. Macrophage Polarization Assay

Human monocytes (THP-1, ATCC) were differentiated into macrophages using PMA/IO (Phorbol 12-myristate 13-acetate/Ionomycin), and, through M1-macrophage generation medium (PromoCell, Heidelberg, Germany), they were polarized to M1 macrophages. A coculture of 5.0 × 10^4^ PMA/IO-stimulated THP-1 (macrophages) and 5 × 10^5^ of TSPCs (n = 3) was performed for 2 days in 24-well plate transwells in M1-macrophage generation medium. Using a polarization qPCR array (ScienCell, Carlsbad, CA, USA), the macrophage polarization status was assessed.

According to the manufacturer’s instructions, RNA was extracted from THP-1 cultures using the RNeasy Mini Kit (Qiagen, Frederick, MD, USA). An amount of 1 μg of total RNA was reverse transcripted with a SuperScript™ VILO™ cDNA synthesis kit (Invitrogen). The cDNA obtained was used to perform a pre-designed 40-gene human macrophage polarization array (GeneQuery™ Human Macrophage Polarization Marker qPCR Array Kit, ScienCell) and processed using a StepOne Real-time thermocycler (Applied Biosystems, LLC, Foster City, CA, USA). GAPDH was used as a housekeeping gene to normalize the mean values obtained, and the expression levels were calculated using the 2^−ΔΔCt^ method. Values were represented in a stacked bar plot for M0, M1, and M2 polarization as the relative fold change of the PMA/IO + THP-1/TSPCs to PMA/IO + THP-1 (reference sample, 2^−ΔΔCt^ = X sample/X reference sample).

### 2.5. Achilles and Patellar Tendinopathy Induction and Tissue Explant Collection

Ten-week-old female (n = 6) and male (n = 6) Sprague Dawley 250 g rats were included in this study. Animals were housed and acclimated for 1 week after arrival, with food and water ad libitum, in sanitary, ventilated rooms with controlled temperature, humidity, and 12 h light/dark cycles. Intratendinous injections were performed on rats anesthetized by isoflurane inhalation. All animals were treated to induce both the Achilles and the patellar tendinopathies in the right limb. Achilles tendinopathy was induced by administering 90 µg of type I collagenase (solubilized in 30 µL of sterile saline solution, 0.9%, with a final concentration of 3 mg/mL) in the medial side of the right Achilles tendon (n = 12) with a 30 G needle without tendon exposure [30]. Similarly, the same rats received 500 ng of prostaglandin E2 (solubilized in 30 µL of sterile saline solution, 0.9%, with a final concentration of 16 µg/mL solution in the right patellar tendons (n = 12) without exposure to induce patellar tendinopathy model [31]). Contralateral left Achilles and patellar tendons were left untreated, representing the control healthy group (n = 4, Achilles; n = 4, patellar tendons), or were injected with the same volume of saline solution, representing the control sham group (n = 8, Achilles; n = 8, patellar tendons). Animals were monitored daily, and free cage activity was permitted after injection. After 7 days, animals were sacrificed, and tendons were harvested for subsequent analysis described below. 

### 2.6. Samples Processing and Histological Analysis

Achilles (n = 24) and patellar tendons (n = 24) were fixed with 10% neutral buffered Formalin for 24 h and then stored in 0.9% saline solution. Samples were dehydrated in an increasing scale of ethanol (70%, 80%, 96%, 100%) before their embedding in paraffin and sectioned using a microtome into 5-micron sections and mounted onto glass slides. Tendon slides were stained with hematoxylin and eosin (H&E) (Carlo Erba) after de-paraffinization and re-hydration. Photomicrographs and image stitching were acquired using an Olympus IX71 optical microscope and an Olympus XC10 camera (Tokyo, Japan).

H&E-stained sections of each rat and condition (pathological, healthy control group, and sham control group) were evaluated blindly by four independent observers. A semi-quantitative evaluation score was prepared based on previous studies [32,33], with some additional evaluations (Table 2). Both the mid portion of the tendon proper and the paratenon were evaluated according to a modified version of Chen et al. score [32,33]. With reference to the tendon mid portion, the parameters evaluated were the structure and arrangement of the fibers, cell morphology and cell density, neovascularization, infiltration of inflammatory cells, and adipose tissue accumulation. With reference to the paratenon, the parameters evaluated were its thickness, neovascularization, infiltration of inflammatory cells, and cell density. Each parameter was quantified using a 0–3 grading scale (Table 2).

### 2.7. Immunofluorescence Analysis

Tendon slides on coverslips were heated in a water bath at 60 °C overnight in antigen retrieval buffer H (ThermoFisher Scientific, Waltham, MA, USA). After washes in PBS 1X (Phosphate saline buffer) and PBS supplemented with Tween 20 (1:2000) (Sigma-Aldrich, Gallarate, MI, Italy), slides were permeabilized for 10 min with Triton X-100 0.01% (Sigma-Aldrich, Gallarate, MI, Italy) diluted in PBS and blocked in 3% bovine serum albumin (BSA) (Sigma-Aldrich, Gallarate, MI, Italy) for 30 min at room temperature. After blocking, multiple immune staining was performed. Slides were incubated overnight at 4 °C with primary antibodies diluted in 1% BSA. Anti-CD146 (ab75769, Abcam, Cambridge, UK) and anti-CD90/THY (ab181469, Abcam), diluted at 1:250 and 1:200, respectively, were used to localize TSPCs within the tissue. Anti-CD86 (ab220188, Abcam) diluted at 1:500 and anti-CD206 (ab64693, Abcam) diluted at 1:500 was used to detect M1 and M2 macrophages, respectively. Secondary antibody incubation was performed using AlexaFluor488 polyclonal anti-rabbit IgG, (H+L) (ab150077) or AlexaFluor647 polyclonal anti-mouse IgG, (H+L) (ab150115, Abcam) secondary antibodies, both raised in goat, diluted at 1:500 in PBS 1% BSA, for 1 h at room temperature. Nuclei were detected with DAPI (10 µg/mL) diluted in PBS 1X for 10 min at room temperature.

Each sample was observed and acquired as both a stitched overlay image (10× and 20× magnification) and a single overlay image (10× and 20×). Fluorescent images were acquired with an Olympus IX71 microscope.

### 2.8. Fluorescence Analysis and Quantification

Fluorescent signals of each marker were quantified with ImageJ software (ImageJ 1.53e). All fluorescence images were pre-processed to adjust brightness and contrast using the same parameters for each channel, thus preserving the differences in fluorescence intensity among the various samples. Fluorescence intensity quantification was performed in defined regions of interest (ROIs) identified by the “freehand area” tool. Three separate ROIs were considered for each sample for both the mid-tendon portion and the paratenon areas. A background area was also defined. Through the “ROI Manager” tool, the defined area, the integrated density, and the mean fluorescence of background readings were measured. These parameters were used to calculate the corrected total cell fluorescence (CTCF) in the selected ROIs. The average of the CTCF values derived from the three ROIs of the same sample was normalized according to DAPI values for every marker.

### 2.9. Colocalization of CD90/CD146 and CD86/CD206 Fluorescence Signals

Colocalization of fluorescence signals for CD90/CD146 and for CD86/CD206 was performed to identify cells that co-express CD90/CD146 and CD86/CD206 markers. The ImageJ plugin “Colocalization” (ImageJ 1.53e) was used to generate an 8-bit image of colocalized point, representing the cells doubly positive for CD146 and CD90 or for CD86 and CD206. Moreover, using the same methods described above, we quantified the co-localized points in the tendon proper and paratenon ROIs, corresponding to TSPCs or macrophages, respectively.

### 2.10. Statistical Analysis

GraphPad Prism Software version 8.0.2 (GraphPad, San Diego, CA, USA) was used to perform statistical analyses. Data distribution was assessed via the Shapiro–Wilk normality test (α of 0.05). In all analyzed data, a normal distribution was not observed and non-parametric Kruskal–Wallis and Dunn’s post hoc tests were carried out. The level of significance was set at *p*  ≤  0.05.

## 3. Results

### 3.1. Morphological and Histological Alterations in Achilles and Patellar Tendinopathies

In the present study, we have characterized the pathological alterations occurring in Achilles and patellar tendinopathies at 7 days following agent injection for creating the tendinopathy. Importantly, we selected the specific time point in accordance with previous studies that marked 4 to 7 days as the period when reparative cells migrate and populate the tendon injury site [34,35]. Structurally, pathological manifestations were observed in both tissues, related to the disorganized morphology of tendon fibers and the increased presence of vasculature and adipocytes embedded within tendon fibers. These alterations were more evident in the Achilles tendon (Figure 1A).

Furthermore, quantitation was performed at both the mid portion of the tendon proper and the paratenon, and both tendons were assigned total and partial histological scores using our scoring system (0–3) for tendon histological evaluation (Table 2). The total score was significantly (*p* < 0.05) higher in the pathological Achilles mid tendon proper group compared to the corresponding healthy group and in the pathological patellar paratenon group compared to the healthy and sham groups. The modified Chen et al. score [32,33] for the Achilles tendon showed significantly (*p* < 0.05) increased cell morphology in the pathological tendon proper group compared to the healthy tendon proper group and increased cell density in the paratenon pathological group compared to both the healthy and sham controls. In comparison, the modified Chen et al. score [32,33] for the patellar tendon proper revealed significantly (*p* < 0.05) increased cell density, average thickness, and inflammatory cell infiltration in the pathological group compared to the sham and healthy groups.

### 3.2. TSPCs Localization and Quantitation in Achilles and Patellar Tendinopathies

TSPCs share several surface antigen receptors with MSCs, supporting the concept of a shared progenitor cell [22]. CD146 expression levels are of particular importance as previous studies showed that CD146^+^ MSCs constitute a bona fide perivascular component of the bone marrow niche [33]. Functionally, CD146 expression is associated with innately higher immunomodulatory and secretory capacity and thus potential therapeutic potency [34]. Herein, we have characterized the cellular phenotype alterations occurring in Achilles and patellar tendinopathies on day 7 following injection of either type I collagenase (Achilles tendon) or prostaglandin E2 (patellar tendon). Interestingly, both tendinopathies showed similar CD146 and CD90 expression levels for healthy, sham, and pathological groups in tendon proper and paratenon areas (Figure 2A, left and middle graphs). However, even though not statistically significant, both tendinopathies showed a high (>1) CD146/CD90 ratio in the pathological group at the paratenon site, indicating the recruitment of CD146^+^ TSPCs.

### 3.3. Macrophage Localization and Quantitation in Achilles and Patellar Tendinopathies

Macrophage polarization was investigated in both tendinopathy models. Interestingly, both tendinopathies showed similar CD86 (M1 macrophage) and CD206 (M2 macrophage) expression levels for healthy, sham, and pathological groups in tendon proper and paratenon areas (Figure 3A, left and middle graphs).

However, even though not statistically significant, both tendinopathies showed a lower CD86/CD206 ratio in the pathological group at the paratenon compared to the tendon proper. This outcome can be related to the high CD146^+^ TSPCs presence in the paratenon. Specifically, we have previously shown that CD146^+^ expression in stem/progenitor cells is not only related to their perivascular in vivo topography [36]; most importantly, it is directly associated with innately higher immunomodulatory and secretory capacity and thus potential therapeutic potency [37].

### 3.4. TSPCs Effects on Macrophages Polarization In Vitro

Immunophenotypic analysis for MSC-defining markers indicated high expression levels for CD105 (98.7 ± 0.7%), CD73 (99.9 ± 0.02%), and CD44 (99.9 ± 0.01%). Importantly, CD146 and CD90 markers used to immunolocalize TSPCs embedded and mobilized within the Achilles and patellar tendons were also highly expressed in the cultured TSPCs in vitro. CD146 showed 91.7 ± 5.3% and CD90 showed 99.8 ± 0.1% expression levels at P3 TSPCs (Figure 4A). Therefore, cultured TSPCs show a phenotypic profile consistent with their in situ topology and mobilization in tendinopathies.

In the present study, we have investigated the effects of TSPCs on macrophage polarization in vitro in order to further clarify if the observed reduction in M1 macrophage polarization in situ is related to a greater abundance of reparative TSPCs. Specifically, upon exposure to TSPCs, PMA/IO-stimulated THP-1 molecular profiling indicated a strong gene expression shift from M1 pro-inflammatory phenotype towards an M2 alternative macrophage polarization (Figure 4B). Most importantly, the expression levels of the *MRC1* (CD206) characteristic M2-polarization macrophage marker [38] was strongly induced when macrophages were exposed to TSPCs.

## 4. Discussion

In the present study, we have characterized the pathological alterations occurring in Achilles and patellar tendinopathies at 7 days following agent injection for creating the tendinopathy. Importantly, we selected the specific time point in accordance with previous studies that marked 4 to 7 days as the period when reparative cells migrate and populate the tendon injury site [34,35]. Overall, scoring system data indicate that tendinopathies affect Achilles and patellar tendons differently, with most pathological alterations occurring at the mid tendon proper and paratenon, respectively. It appears that the prostaglandin E2 approach used to generate the patellar tendinopathy model induced a stronger inflammatory signaling response, with increased tissue thickness and inflammatory cell infiltration, compared to the Achilles tendinopathy model.

In general, tendinopathies are defined as a condition of tendon non-healing, whereby a chronic dysregulation of local homeostasis is established due to an amplification of the tissue’s inflammatory and immune components, resulting in alteration in the ECM [14]. Earlier studies described tendinopathies as a three-stage process: injury, failed healing response, and clinical presentation [39]. In the first stage, the interaction between the tendon injury and an unfavorable mechanical environment is the catalyst for the pathological pro-inflammatory process. In the second stage, normal tendon healing is diverted due to the unfavorable mechanical environment, the disturbance of local inflammatory responses, and oxidative stress. In the third stage, pathological alterations occur in the ECM and vascularity, cellularity, and cell phenotypes, cytokine profiles, and peritendinous innervation [39].

Interestingly, CD146^+^ TSPCs delineate an interfascicular, likely perivascular cell subpopulation that is recruited in tendon injury via its ligand laminin-α4, thereby promoting endogenous tendon regeneration [34,39]. On this basis, previous studies have shown that improved tendon healing can be achieved via supplementation with growth factors, including connective tissue growth factor (CTGF), platelet-derived growth factor BB (PDGFBB), basic fibroblast growth factor (bFGF), and insulin-like growth factor-1 (IGF1) [40]. In particular, CTGF stimulates CD146^+^ TSPC recruitment, proliferation, and tenogenic differentiation, and, in parallel, results in downregulation of osteo-, chondro-, and adipogenic differentiation programming [24,41,42,43]. Collectively, all previous studies underscore the significance of the CD146^+^ TSPC subpopulation in tendon healing and regeneration. A recent study demonstrated that the interfascicular matrix is a unique tendon cell niche, containing a vascular-rich network, CD31^+^ endothelial cells, Desmin^+^ mural cells, and CD146^+^ cell populations that are likely essential to tendon healing [34,35]. Coupling our data to previous studies, we can conclude that CD146^+^ TSPCs are recruited upon injury from the interfascicular tendon matrix to the paratenon and thus likely participate in tendon healing even 7 days post-injury. Interestingly, this recruitment was evident independently of the type of tendinopathy, indicating a common mechanism for CD146^+^ TSPC-based tendon healing.

Resident immune-sensing cells respond promptly to the initial tendon insult via damage-associated molecular patterns (DAMPs) and, in conjunction with aberrant tenocyte activation, contribute to the recruitment of infiltrating immune cells to the damaged area. Tenocytes release endogenous agents, that together with the inflammatory cytokines (TNF-α, IL-1β, IL-6, IL-8) and prostaglandins (PGE2) activated by the infiltrating immune cells, promote pro-inflammatory macrophage (M1) polarization to the M2 alternative phenotype [12,18,19]. For these reasons, when immune cell concentration reaches a critical threshold, the release of cytokines has a profound effect on the balance between reparative and degenerative processes, leading to a state of inflammation. In the present study, the observed reduction in M1 macrophage polarization in situ is related to a greater abundance of reparative CD146^+^ TSPCs.

Culture-expanded TSPCs are generally defined as clonogenic, self-renewing, and multipotent cells, expressing a surface antigen profile shared with mesenchymal stem/stromal cells (MSC), e.g., CD44^+^, CD90^+^, CD105^+^, CD146^+^, CD31^−^, and CD45^−^ [21,25,44]. According to our previous in vitro studies, TSPCs show immunomodulatory capacity to suppress stimulated T cells and the ability to degrade the nociceptive stimulator substance P that is produced in the early phases of tendinopathy [45,46]. Herein, culture-expanded TSPCs effectively polarized M1 pro-inflammatory macrophages towards an M2 alternative phenotype. Collectively, our data show that TSPCs presence, both in vitro and in vivo, is directly related to M2 macrophages polarization, therefore attenuating the effects on pro-inflammatory signaling.

Since this is a preliminary study, the main limitation is the low number of TSPC donors (n = 6) and of rats employed (n = 12), and it is necessary to carry out further studies by expanding the number of samples to be analyzed. Furthermore, in future studies, rat-derived TSPCs and macrophages will be combined in co-cultures for a macrophage polarization assay.

Taken together, this information is preliminary for future studies that will provide clinicians with effective and innovative therapeutic approaches for the treatment of tendinopathies.

## 5. Conclusions

The results of this study improve our knowledge about the structural and cellular tendon alterations in both Achilles and patellar tendinopathies, highlighting the significance of TSPCs in immunomodulating the tendon microenvironment. Our findings confirm that upon tendon insult, CD146^+^ TSPCs are recruited to the paratenon area and appear to immunoregulate the tissue microenvironment via polarization of M1 pro-inflammatory macrophages to the M2 phenotype. Although preliminary, this evidence suggests that given the regenerative potential of CD146^+^ TSPCs, they represent a key phenotype to be explored and exploited for the development of targeted regenerative medicine therapies for tendon disorders.

## Figures and Tables

**Figure 1 biomedicines-12-00995-f001:**
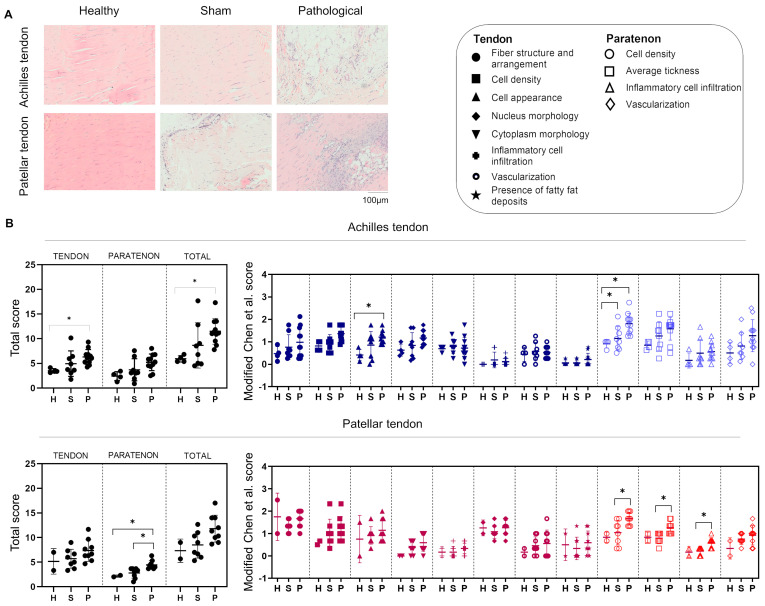
Tendon histological scoring. (**A**) Hematoxylin/eosin staining was performed on histological sections from healthy, sham, and pathological groups of both Achilles and patellar tendons. In both Achilles and patellar heathy sections, the tendon fibers appear aligned and well organized, with evenly distributed elongated cells. In sham sections, the fibers appear less organized and misaligned. In the sham patellar tendon, there is greater cellularity in the paratenon portion. In the pathological sections, the fibers appear completely disorganized, with blood vessels and adipocytes being particularly abundant in the Achilles section. (**B**) Quantification of histological staining. Achilles and patellar tendons were assigned total and modified Chen et al. histological scores [32,33] as reported in Table 1. H = healthy; S = sham; P = pathologic. The total score is higher in the pathological Achilles tendon group compared to the healthy group (* = *p* < 0.05) and in the pathological paratenon patellar group compared to the healthy and sham groups (* = *p* < 0.05). The modified Chen et al. score [32,33] of the Achilles tendon showed higher values of cell appearance in the pathological tendon group compared to the healthy tendon group (* = p < 0.05) and of cell density in the paratenon pathological group compared to the healthy and sham groups (* = *p* < 0.05). The modified Chen et al. score of the patellar tendon showed higher values for cell density, average thickness, and inflammatory cell infiltration in the pathological group compared to the sham and healthy groups (* = *p* < 0.05). Histological score data are reported as medians with interquartile ranges. For all these data, non-parametric Kruskal–Wallis and Dunn’s post hoc tests were carried out.

**Figure 2 biomedicines-12-00995-f002:**
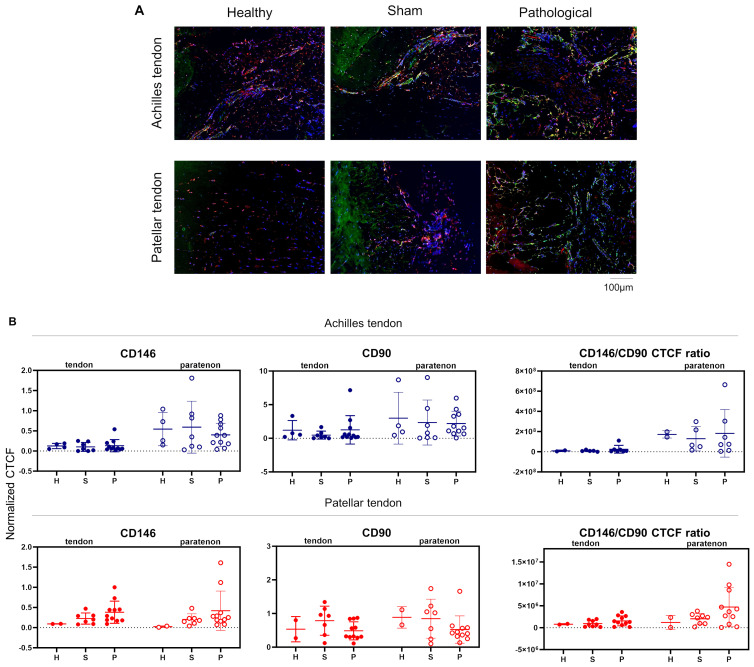
Immunolocalization of CD90^+^ CD146^+^ TSPCs. (**A**,**B**) Immunofluorescence was performed for CD90 (red), CD146 (green), and DAPI (nuclei, blue) on histological sections from healthy, sham, and pathological groups of both Achilles and patellar tendons. H = healthy; S = sham; P = pathological. Both tendinopathies showed a high (>1) CD146/CD90 ratio in the pathological group at the paratenon site, indicating the recruitment of CD146^+^ TSPCs. For all these data, normality testing was not observed; therefore, a non-parametric Kruskal–Wallis and Dunn’s post hoc tests were carried out.

**Figure 3 biomedicines-12-00995-f003:**
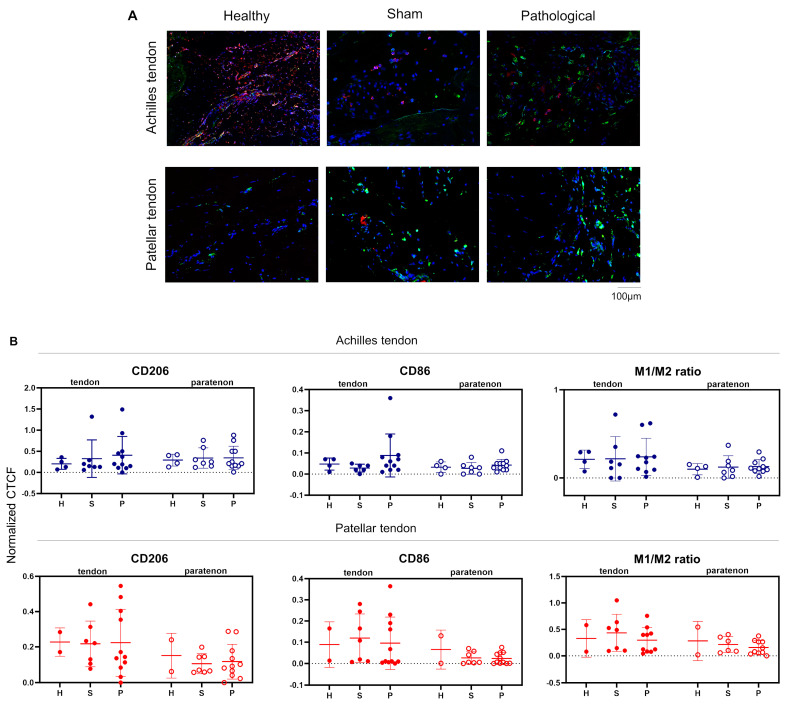
Immunolocalization of CD86 and CD206 macrophages. (**A**,**B**) Immunofluorescence was performed for CD86 (red), CD206 (green), and DAPI (nuclei, blue) on histological sections from healthy, sham, and pathological groups of both Achilles and patellar tendons. H = healthy; S = sham; P = pathological. Both tendinopathies showed a lower CD86/CD206 ratio in the pathological group at the paratenon compared to the tendon proper. For all these data, normality testing was not observed; therefore, non-parametric Kruskal-Wallis and Dunn’s post hoc tests were carried out.

**Figure 4 biomedicines-12-00995-f004:**
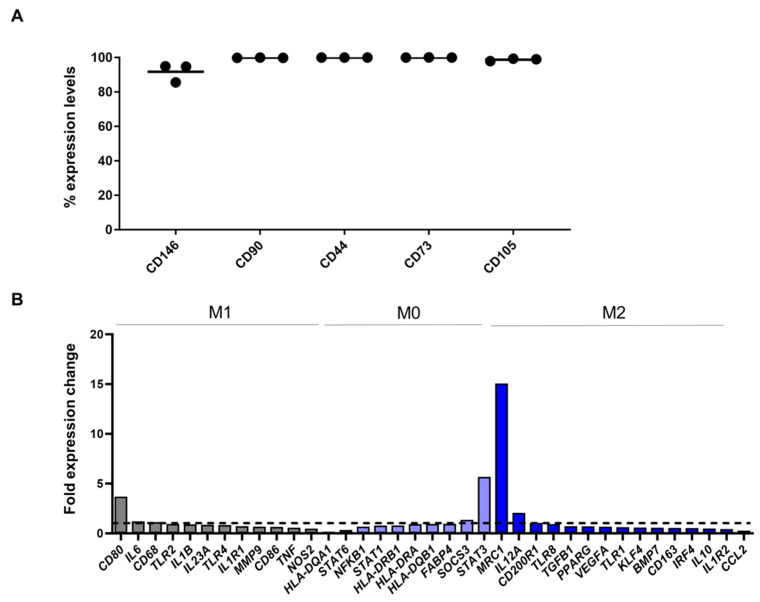
TSPCs immunophenotypic profiling and co-culturing with macrophages in vitro. (**A**) Cultured TSPCs showed high expression (>90%) of MSC-defining markers (CD105, CD73, CD44) and CD146 and CD90 markers used to immunolocalize TSPCs embedded within the Achilles and patellar tendons. (**B**) Upon exposure to TSPCs, PMA/IO-stimulated THP-1 gene expression analysis showed a strong shift towards M0/M2 macrophage polarization. Notably, exposure to TSPCs induced significant expression of the key M2-polarization marker MRC1. M1 (grey bars); M0 (light blue bars); M2 (blues bars). Black dots indicated the threshold of the shift towards M0/M2 macrophage polarization.

**Table 1 biomedicines-12-00995-t001:** Demographic data of donors.

	Gender	Age (years)	Weight (kg)	Height (cm)	BMI (kg/m^2^)
Patient 1	M	42	65	172	21.97
Patient 2	M	24	70	178	22.09
Patient 3	M	31	78	180	24.07
Patient 4	M	42	82	175	26.78
Patient 5	M	34	79	185	23.08
Patient 6	M	23	87	192	23.60

**Table 2 biomedicines-12-00995-t002:** Modified Chen et al. histological score of tendon mid-portion and paratenon portion.

Tendon Mid-Portion					
**Structure and arrangement of fibers**	**0**	Continuous, parallel collagen fibers	**Cytoplasm morphology**	**0**	No obvious cytoplasm
	**1**	Partially disorganized and fragmented fibers		**1**	Slightly increased cytoplasm
	**2**	Moderately disorganized, fragmented, crossed and wavy fibers		**2**	Moderate cytoplasm
	**3**	Total disorganized and non-identifiable fiber pattern		**3**	Abundant cytoplasm
**Cell density**	**0**	Normal	**Infiltration of inflammatory cells**	**0**	<10%
	**1**	Slightly increased		**1**	10–20%
	**2**	Moderately increased		**2**	20–30%
	**3**	Markedly increased		**3**	>30%
**Cell appearance**	**0**	Spindle-shape cells	**Neovascularization**	**0**	No blood vessels
	**1**	Slightly rounded cells		**1**	Slight increase of vascular bundles
	**2**	Moderately rounded cells		**2**	Moderate increase of vascular bundles
	**3**	Markedly rounded cells		**3**	Marked increase of vascular bundles
**Nucleus morphology**	**0**	Thin and elongated	**Fatty deposits**	**0**	Absence of lipid vacuoles
	**1**	Slightly round nucleus		**1**	Slight increase of lipid vacuoles
	**2**	Round nucleus		**2**	Moderate increase of lipid vacuoles
	**3**	Large and round nucleus		**3**	Marked increase of lipid vacuoles
**Paratenon**					
**Cell density**	**0**	Normal	**Inflitration of inflammatory cells**	**0**	<10%
	**1**	Slightly increased		**1**	10–20%
	**2**	Moderately increased		**2**	20–30%
	**3**	Markedly increased		**3**	>30%
**Average thickness**	**0**	Normal	**Neovascularization**	**0**	Normal presence of vascular bundles
	**1**	Slightly increased		**1**	Slight increase of vascular bundles
	**2**	Moderately increased		**2**	Moderate increase of vascular bundles
	**3**	Markedly increased		**3**	Marked increase of vascular bundles

## Data Availability

Raw data for this study are available at https://osf.io/bw5e8/ (accessed on 1 March 2024).

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
