# Peer review of "Cellular and Structural Changes in Achilles and Patellar Tendinopathies: A Pilot In Vivo Study"

_biomedicines, 2024, doi:10.3390/biomedicines12050995_

Round 1

Reviewer 1 Report

Comments and Suggestions for Authors Interesting article on a little covered topic. The topic of the manuscript is Cellular and Structural Changes in Achilles and Patellar Tendinopathies. There are few related articles. My comments: Abstract: It should be added that this is a pilot study. The conclusions are based on a small number of people - only 6. Introduction: Please increase the paragraph and literature review regarding the epidemiology and etiology of Achilles and Patellar Tendinopathies Please ad research hypothesis. Materials and methods: clearly and extensively presented. Small study group – only 6 people. Please add patient demographics (race, weight, height, BMI......) Good description of the research methodology. Well-selected statistical tools. Clear and good quality figures. Results: Well written. Well and precisely presented research results. Clear, legible and good quality tables and figures. The discussions: well written, extensive literature review. Please add work restrictions. It should be clearly stated that the conclusions are based on 6 cases, that these are preliminary studies and that similar studies on a larger number of patients are necessary. Practical conclusions for orthopedists should be added, which result from this work. What possibilities do the results of this work offer in practice? Conclusions: Practical conclusions for orthopedists should be added, which result from this work. What possibilities do the results of this work offer in practice? Review of literature related to the topic should be increased, especially regarding the global epidemiology and etiology of Achilles and patellar Tendinopathies.

Author Response

I thank the reviewer for the valuable comments. As suggested we added that this is a pilot study in the abstract section (line 26).We also included the limitations in the discussion (lines 430-432), represented by the low number of subjects (for in vitro experiments) and rats (for in vivo experiments). As recommended we have included a table with the demographic data of the six TSPC donor patients. In addition, this pilot study lays the foundation for subsequent studies in which TSPCs can be exploited as a therapeutic target to regenerate damaged tendon tissue. This will provide clinicians with additional and effective tools in the treatment of tendinopathies.As suggested, this concept has now been added into the text (lines 433-435).

The literature has been implemented.

Reviewer 2 Report

Comments and Suggestions for Authors

Cellular and Structural Changes in Achilles and Patellar Tendinopathies: A Pilot In Vivo Study

The authors aimed to assess whether tendon stem/progenitor cells (TSPCs) could have a role in enhancing tendon repair.

They characterized the structural and cellular alterations that occurred 7 days after tendon damage in Achilles and Patellar tendinopathy models.

After injury, CD146+ TSPCs were recruited from the interfascicular tendon matrix to the vicinity of the paratenon. There was an observed reduction in M1 macrophage polarization when there was a greater presence of reparative CD146+ TSPCs in situ.

TSPCs also appeared to demonstrate similar immunomodulatory effects on macrophages in vitro, where the TSPCs effectively polarized M1 macrophages towards an anti-inflammatory therapeutic M2 phenotype.

This study sheds some light on the structural and cellular tendon alterations in both Achilles and Patellar tendinopathies, highlighting the significance of TSPCs in immunomodulating the tendon microenvironment. The authors concluded that CD146+ TSPCs were a key phenotype that warranted further investigation in the development of targeted regenerative therapies for tendinopathies.

I think this is a very good and thorough study. Though the cellular mechanisms are not entirely clear, this should spark some interest in further exploring these TSPCs. This is good news for the many sufferers of tendinopathies!

Overall, I did not feel that any significant changes were required and I am happy for this to be published following some language editing.

Comments on the Quality of English Language

Some English language editing is required

Author Response

We sincerely thank the reviewer for appreciating this paper.

Reviewer 3 Report

Comments and Suggestions for Authors

Dear Authors,

I would like to congratulate you on your manuscript.

The purpose of your article is extremely interesting, since we have to fully understand the biology behind tendinopathy before being able to give definitive recommendations for its treatment.

The overall quality of the manuscript is excellent in all its part, especially regarding the stem/progenitor cells isolation and samples preparation, processing and histological analysis.

I do not have any particular suggestion for you, I may just recommend to cite the following articles:

- Tarantino D, Mottola R, Resta G, Gnasso R, Palermi S, Corrado B, Sirico F, Ruosi C, Aicale R. Achilles Tendinopathy Pathogenesis and Management: A Narrative Review. Int J Environ Res Public Health. 2023 Aug 30;20(17):6681. doi: 10.3390/ijerph20176681. PMID: 37681821; PMCID: PMC10487940.

- Aicale, R., Tarantino, D., Maffulli, N. (2017). Basic Science of Tendons. In: Gobbi, A., Espregueira-Mendes, J., Lane, J., Karahan, M. (eds) Bio-orthopaedics. Springer, Berlin, Heidelberg. https://doi.org/10.1007/978-3-662-54181-4_21

Finally, there is no "strengths and limitation" sub-paragraph. The points of strength were already reported, so I may also suggest to report the limitations (if any)

Author Response

The authors thank the reviewer for the positive comments and for suggesting the inclusion of limitation. Although this was not included as a subparagraph, it is found within the discussion section (lines 430-432).

Reviewer 4 Report

Comments and Suggestions for Authors

Dear authors, this is a very important study, but during the review process as a reviewer I found two separated experiments. It is necessary to re-write some paragraphs in the introduction section and M&M sections to let to the readers to understand better the nature and the aims of your study. In the section of the statistical analysis you must mention that variables were analyzed by parametric tests and which variables were evaluated by non-parametric tests. The results of the histology analysis  should be evaluated by non-parametric tests because they are discrete categorical variables. These data could also be evaluated by using a GLMM using a Poisson family. I Aldo suggest to change the figures of the histology score for tables indicating the median and IQ range values.

Author Response

We thank the reviewer for the valuable suggestions.

In order tounderstand better the nature and the purpose of our study we have explicated the aims (lines 89-92). About the statistic part, all histological data were analysed as non-parametric tests as also shown by the raw data deposited on the repository file (https://osf.io/bw5e8/files/osfstorage?view_only=).This specification has now been included in the text. The evaluation of histological score data was performed according to Chen et al. In addition, as suggested, we changed the figures of the histology score indicating the median and IQ range values.

Round 2

Reviewer 1 Report

Comments and Suggestions for Authors The authors introduced the suggested changes and the manuscript is accepted in its current form.

Author Response

The authors thank the reviewer for the valuable suggestions.

Reviewer 4 Report

Comments and Suggestions for Authors

All my concerns were addressed.

Author Response

(The authors gave the same response as above.)
